# The Xanthine Oxidase Inhibitor Febuxostat Suppresses Adipogenesis and Activates Nrf2

**DOI:** 10.3390/antiox12010133

**Published:** 2023-01-05

**Authors:** Yoshiki Higa, Masahiro Hiasa, Hirofumi Tenshin, Emiko Nakaue, Mariko Tanaka, Sooha Kim, Motosumi Nakagawa, So Shimizu, Kotaro Tanimoto, Jumpei Teramachi, Takeshi Harada, Asuka Oda, Masahiro Oura, Kimiko Sogabe, Tomoyo Hara, Ryohei Sumitani, Tomoko Maruhashi, Hiroki Yamagami, Itsuro Endo, Toshio Matsumoto, Eiji Tanaka, Masahiro Abe

**Affiliations:** 1Department of Orthodontics and Dentofacial Orthopedics, Graduate School of Biomedical Sciences, Tokushima University, 3-18-15 Kuramoto, Tokushima 770-8503, Japan; 2Department of Hematology, Endocrinology and Metabolism, Graduate School of Biomedical Sciences, Tokushima University, 3-18-15 Kuramoto, Tokushima 770-8503, Japan; 3Department of Oral Function and Anatomy, Graduate School of Medicine Dentistry and Pharmaceutical Sciences, Okayama University, 2-5-1 Shikata-cho, Kita-ku, Okayama 700-8530, Japan; 4Department of Bioregulatory Sciences, Graduate School of Medical Sciences, Tokushima University, 3-18-15 Kuramoto, Tokushima 770-8503, Japan; 5Fujii Memorial Institute of Medical Sciences, Tokushima University, 3-18-15 Kuramoto, Tokushima 770-8503, Japan

**Keywords:** obesity, adipocytic differentiation, reactive oxygen species (ROS), xanthine oxidoreductase (XOR), febuxostat, Nrf2, Keap1

## Abstract

Xanthine oxidoreductase (XOR) is a rate-limiting enzyme in purine catabolism that acts as a novel regulator of adipogenesis. In pathological states, xanthine oxidoreductase activity increases to produce excess reactive oxygen species (ROS). The nuclear factor erythroid 2-related factor 2 (Nrf2) is a critical inducer of antioxidants, which is bound and repressed by a kelch-like ECH-associated protein 1 (Keap1) in the cytoplasm. The Keap1-Nrf2 axis appears to be a major mechanism for robust inducible antioxidant defenses. Here, we demonstrate that febuxostat, a xanthine oxidase inhibitor, alleviates the increase in adipose tissue mass in obese mouse models with a high-fat diet or ovariectomy. Febuxostat disrupts in vitro adipocytic differentiation in adipogenic media. Adipocytes appeared at day 7 in absence or presence of febuxostat were 160.8 ± 21.2 vs. 52.5 ± 12.7 (*p* < 0.01) in 3T3–L1 cells, and 126.0 ± 18.7 vs. 55.3 ± 13.4 (*p* < 0.01) in 10T1/2 cells, respectively. Adipocyte differentiation was further enhanced by the addition of hydrogen peroxide, which was also suppressed by febuxostat. Interestingly, febuxostat, but not allopurinol (another xanthine oxidase inhibitor), rapidly induced the nuclear translocation of Nrf2 and facilitated the degradation of Keap1, similar to the electrophilic Nrf2 activator omaveloxolone. These results suggest that febuxostat alleviates adipogenesis under oxidative conditions, at least in part by suppressing ROS production and Nrf2 activation. Regulation of adipocytic differentiation by febuxostat is expected to inhibit obesity due to menopause or overeating.

## 1. Introduction

Obesity is defined as an increased adipose mass tissue with hyperplasia and hypertrophy of adipocytes [1,2] and has become a major health issue in the modern era [3,4,5]. An accumulation of visceral fat plays a major role in the pathogenesis of the metabolic syndrome [6,7,8,9,10]. Fat accumulation is correlated with systemic oxidative stress, and excessive adiposity can accelerate the formation of toxic free radical species [11,12]. Reactive oxygen species (ROS) are produced in cells to regulate cellular functions under physiological and pathological conditions [13,14,15,16]. Xanthine oxidoreductase (XOR) is a rate-limiting enzyme in purine catabolism that catalyzes hypoxanthine from nucleic acid metabolites into xanthine and xanthine into uric acid. ROS are generated by these processes [17]. In pathological states, XOR activity increases to produce excessive ROS with attendant effects.

Febuxostat is a xanthine oxidase (XO) inhibitor that has been used in clinical practice for the treatment of hyperuricemia and gout [18]. Febuxostat effectively inhibits ROS formation [19]. We previously reported that febuxostat inhibits the receptor activator of NFκB ligand (RANKL)-mediated osteoclastogenesis by suppressing ROS production in osteoclast progenitors and alleviating bone loss in ovariectomized mice [20]. The anticancer agent doxorubicin enhanced RANKL-induced ROS production and osteoclastogenesis, which were abolished by febuxostat. These results suggest that XOR inhibition by febuxostat can ameliorate XOR-mediated ROS overproduction under pathological conditions. In addition, the antioxidant N-acetylcysteine (NAC) suppresses adipocyte differentiation by decreasing ROS [21,22,23], suggesting a critical role for ROS in adipocyte differentiation [24].

Nuclear factor erythroid 2-related factor 2 (Nrf2) is a critical regulator of oxidative stress. Under normal cellular conditions without excessive oxidative stress, the kelch-like ECH-associated protein 1 (Keap1) binds to Nrf2 in the cytoplasm and ubiquitinates and degrades Nrf2 [25]. However, under excessive oxidative stress, Keap1 releases Nrf2, which allows the translocation of Nrf2 into the nucleus [26]. Nrf2 then binds to an antioxidant response element (ARE) and upregulates the transcription of its target antioxidant genes to protect cells from excessive oxidative stress [27]. Therefore, the Keap1-Nrf2 axis appears to be a major mechanism for robust inducible antioxidant defense against various exogenous insults [28].

In the present study, we explored the effects of febuxostat on adipocyte differentiation by focusing on the regulation of ROS via the Keap1-Nrf2 axis. Here, we demonstrate that febuxostat ameliorates obesity in mice with a high-fat diet or ovariectomy and that it suppresses adipocyte differentiation in adipogenic media. Furthermore, it facilitates the degradation of Keap1, thereby inducing nuclear translocation of Nrf2 to induce the expression of its target antioxidant factors, such as heme oxygenase-1 (HO-1) and NAD(P)H quinone dehydrogenase 1 (NQO1). These results suggest that febuxostat acts as an Nrf2 activator and suppresses ROS production through XO inhibition to alleviate the oxidative condition of adipose progenitor cells. Febuxostat may be a potential agent for modulating excessive adipocyte differentiation in obesity.

## 2. Materials and Methods

### 2.1. Reagents

The following reagents were purchased from the indicated manufacturers: rabbit antibodies against Pparγ, C/ebpα, mouse antibody against β-actin, horseradish peroxidase (HRP)-conjugated anti-rabbit and anti-mouse IgG from Cell Signaling Technology (Beverly, MA, USA); rabbit antibodies against XO and Keap1 and mouse antibody against Nmp84 from Abcam (Dallas, TX, USA); rabbit antibody against Nrf2 from GeneTex (Irvine, CA, USA); recombinant human insulin, 3-isobutyl-1-methyl xanthine (isobutylmethylxanthine, IBMX), dexamethasone, febuxostat, allopurinol, cycloheximide, and troglitazone from Wako Pure Chemicals Corporation (Osaka, Japan); NAC from Nacalai Tesque (Kyoto, Japan); and omaveloxolone from MedChemExpress (Monmouth, NJ, USA).

### 2.2. Animal Experiments

Animal experiments were performed with the permission of the Animal Care and Use Committee of Tokushima University, Tokushima, Japan (T2020-110). Female 8-week-old C57BL/6J mice were purchased from Charles River Laboratories Japan (Tokyo, Japan). The mice were randomly divided into four groups (*n* = 5 per group): regular diet groups with or without febuxostat administration and high-fat diet groups with or without febuxostat administration. Febuxostat was administered daily by oral gavage at 5 mg/kg for 10 weeks. The animals were weighed weekly. Visceral white adipose tissues were collected at the end of the study, fixed with 10% formalin (Wako Pure Chemicals Corporation), embedded in paraffin, and stained with hematoxylin and eosin (H&E). Quantitative analysis of abdominal fat volume was performed and quantified at 10 weeks from micro-CT images of the proximal end of L1 vertebral level to the distal end of L5 using LaTheata LCT-200 (Hitachi Aloka, Tokyo, Japan).

### 2.3. Cell Cultures

The mouse pre-adipocytic cell line 3T3-L1 and the pre-osteoblastic cell line MC3T3-E1 were obtained from ATCC (Washington, DC, USA), while the mesenchymal cell line C3H10T1/2 was obtained from the RIKEN BioResource Research Center (Tsukuba, Japan). These cells were cultured in Eagle’s minimal essential medium, α modification (Sigma-Aldrich, St. Louis, MO, USA) supplemented with 10% FBS, L-glutamine, and 50 mg/mL penicillin/streptomycin (Thermo Fisher Scientific, Waltham, MA, USA). Adipocytic differentiation was induced using a differentiation cocktail (10 μg/mL insulin, 0.5 mM IBMX, and 0.4 mg/mL dexamethasone). The culture media were changed to medium containing 10 μg/mL insulin 48 h later, and the cells were further cultured until day 7. Febuxostat, NAC (5 mM), allopurinol (50 μM), or omaveloxolone (50 nM) were added to the differentiation cocktail from day 0 and throughout the experimental period. In oxidative stress experiments, hydrogen peroxide was added to the adipocyte differentiation medium, and the cells were treated for 48 h in the hydrogen peroxide-containing medium, according to the previous report [29]. The cells were then washed and further cultured for 7 days in the adipocyte differentiation medium without hydrogen peroxide. The cells were then washed, fixed with 10% neutral-buffered formalin, and stained for intracellular lipid inclusion bodies with 0.5% Oil Red O (Sigma-Aldrich) in isopropanol:distilled water (60:40) for 30 min. Images were obtained using a fluorescence microscope (BZ-X800; Keyence, Osaka, Japan).

### 2.4. Nrf2 Gene Silencing

3T3-L1 and C3H10T1/2 cells were suspended in Opti-MEM (11058-021; Life Technologies, Carlsbad, CA, USA) and transfected with polybrene for 20 h using MISSION^®^ Lentiviral Transduction Particles (Sigma-Aldrich) for Nrf2 shRNA, containing five different Nfr2 shRNA lentiviral constructs and a control shRNA construct.

### 2.5. Cell Viability

Cell viability was determined using the Cell Counting Kit-8 assay (Dojindo, Kumamoto, Japan), according to the manufacturer’s instructions. 3T3-L1 and C3H10T1/2 cells were cultured in 96-well plates with febuxostat for 48 h and then incubated with 2-(2-methoxy-4-nitrophenyl)-3-(4-nitrophenyl)-5-(2,4-disulphophenyl)-2*H*-tetrazolium monosodium salt for 3 h. The absorbance of each well was measured at 450–655 nm by using an iMark Microplate Absorbance Reader (Bio-Rad Laboratories, Hercules, CA, USA).

### 2.6. Real-Time PCR

Total RNA was extracted using the TRIzol reagent (Gibco-BRL, Rockville, MD, USA). Total RNA (2 μg) was reverse-transcribed using PrimeScript RT Reagent Kit (Takara Bio Inc., Shiga, Japan) in a 20 μL reaction solution. To perform real-time PCR, each cDNA sample was amplified using the SYBR^®^ Green Premix Ex Taq^TM^ II kit (Takara Bio Inc.) with a 7300 Real-time PCR System (Thermo Fisher Scientific). The reaction conditions consisted of 2 μL cDNA and 0.4 μM primers in a total volume of 20 μL. *Gapdh* was used as an endogenous control to normalize each sample. The following primer sequences were used: mouse *Ap2* F: GATGCCTTTGTGGGAACCTG and R: GAATTCCACGCCCAGTTTGA, mouse *C/ebpα* F: TGGACAAGAACAGCAACGAG and R: TCACTGGTCAACTCCAGCAC, mouse *Ho-1* F: CTGTGAACTCTGTCCAATG and R: AACTGTGTCAGGTATCTCC, mouse *Nqo1* F: CATTCTGAAAGGCTGGTTTGA and R: TTTCTTCCATCCTTCCAGGAT, and mouse *Gapdh* F: AAATGGTGAAGGTCGGTGTG and R: TGAAGGGGTCGTTGATGG.

### 2.7. Western Blot Analysis

The cells were collected and lysed using RIPA lysis buffer. To collect nuclear extracts, cells were lysed using NE-PER™ Nuclear and Cytoplasmic Extraction Reagents (Life Technologies), according to the manufacturer’s protocol. The protein concentration in each sample was measured using the Pierce™ BCA Protein Assay Kit (Bio-Rad) and then adjusted. The samples were heated at 95 °C for 5 min. The cell lysates were separated by sodium dodecyl sulfate polyacrylamide gel electrophoresis on a 10% polyacrylamide gel and transferred to polyvinylidene difluoride membranes (Millipore, Billerica, MA, USA). The membranes were then blocked with 4% skim milk and incubated with primary antibodies overnight. After washing, membranes were incubated with the corresponding secondary antibodies. The membranes were analyzed using an Amersham Imager 600 (GE Life Sciences, Little Chalfont, UK). The band intensities from each blot were quantified using the ImageJ software version 2.1.0 (NIH, Bethesda, MD, USA) and normalized to β-actin to calculate fold changes from controls.

### 2.8. Statistical Analysis

First, the data distributions were analyzed using the x^2^ test. Then, statistical differences were assessed using Student’s *t*-test for two normally distributed groups and one-way analysis of variance (ANOVA) with Tukey’s test for multiple normally distributed groups. Mann–Whitney U tests were used for two non-normally distributed groups and Steel–Dwass tests were used for multiple non-normally distributed groups using the Statcel 4 software (OMS Publishing, Saitama, Japan). *p* < 0.05 was considered statistically significant.

## 3. Results

### 3.1. Febuxostat Alleviates Obesity in High-Fat Diet-Fed or Ovariectomized Mice

To examine the effects of febuxostat on obesity, febuxostat was orally administered to obese mice who were fed an obesogenic, high-fat diet. Mice fed a high-fat diet showed a greater accumulation of visceral and subcutaneous adipose tissues along with more weight gain than those fed a normal diet (Figure 1A–C). The daily administration of febuxostat (5 mg/kg) significantly reduced the abdominal fat accumulation area, as determined by cross-sectional CT images, and suppressed weight gain in the mice that were fed a high-fat diet. At 10 weeks, weight gain from the baseline was 55.5 ± 9.1% vs. 10.7 ± 4.5% (*p* < 0.01) in mice fed a high-fat diet and a normal-chow diet, respectively. Histopathological analysis revealed that the size of individual adipocytes was markedly increased in omental fat tissues from the mice that were fed a high-fat diet, which was reduced by the administration of febuxostat (Figure 1D,E). We further examined the effects of febuxostat on obesity after ovariectomy. Ovariectomized mice showed an increase in body weight and abdominal fat compared with the control mice (Figure 2A–E). However, daily oral administration of 5 mg/kg febuxostat prevented the increase in body weight and fat mass in the ovariectomized mice. Interestingly, treatment with febuxostat did not affect the area of abdominal adipose tissue or body weight gain in mice fed a normal diet. These results collectively suggest that febuxostat mitigates the obesogenic effects of a high-fat diet or ovariectomy.

### 3.2. Febuxostat Inhibits Adipocyte Differentiation in Adipogenic Cultures

Next, we examined the effect of febuxostat on adipocyte differentiation and lipid accumulation in vitro. 3T3-L1 and C3H10T1/2 cells were cultured in an adipogenic medium supplemented with IBMX, a commonly used adipogenic inducer. 3T3-L1 cells differentiated in the adipogenic medium into mature adipocytes with lipid droplets, as determined by Oil Red O staining (Figure 3A). However, 20–100 μM febuxostat suppressed differentiation into mature adipocytes in a dose-dependent manner. Both 3T3-L1 and C3H10T1/2 cells differentiated into mature adipocytes with lipid droplets, as shown by the microscopic images taken seven days after the initiation of adipocyte differentiation (Figure 3B). Febuxostat (50 μM) suppressed the differentiation of both cell types into mature adipocytes with lipid droplets. The viability was not affected in 3T3-L1 and C3H10T1/2 cells treated with febuxostat at concentrations up to 100 μM (Figure 3D). Therefore, we conducted the following experiments using 50 μM of febuxostat. A previous report on *Xor* gene silencing in 3T3-L1 cells demonstrated that XOR acts downstream of C/EBPβ to express the transcription factors C/EBPα and PPARγ for further adipocyte maturation [30]. Consistently, febuxostat abolished these adipogenic transcription factors at the protein level in the adipogenic culture of 3T3-L1 cells (Figure 3E). In addition to *C/ebpα*, the expression of ap2, a downstream target of PPARγ and a terminal marker of adipocyte differentiation, was not induced in 3T3-L1 and C3H10T1/2 cells in adipogenic cultures in the presence of febuxostat (Figure 3F). Together with the in vivo effects of febuxostat (Figure 1 and Figure 2), it is suggested that febuxostat suppresses exogenously induced adipogenesis.

### 3.3. ROS Is Vital for Adipocyte Differentiation

ROS produced in cells play an important role under physiological and pathological conditions in response to exogenous stimuli [13,14,15]. XO is a major ROS generator in the cells. Since the XO inhibitor and febuxostat abolished the induction of adipogenesis, we first confirmed the expression of XO during adipogenic induction. The expression of *Xor* mRNA (Figure 4A) and XO protein (Figure 4B) was upregulated in 3T3-L1 and C3H10T1/2 cells upon adipogenic induction. Next, we investigated the involvement of ROS in adipocytic differentiation using NAC, an ROS scavenger. Consistent with previous reports [21,22], the addition of 5 mM NAC almost completely suppressed the appearance of Oil Red O-positive lipid droplets in 3T3-L1 cells and partly in C3H10T1/2 cells (Figure 4C,D), suggesting a critical role for ROS in adipocytic differentiation. Conversely, the addition of hydrogen peroxide further promoted adipocyte differentiation in 3T3-L1 cells upon stimulation with adipogenic media in a dose-dependent manner (Figure 4E). Interestingly, the enhancement of adipocytic differentiation by 10 μM hydrogen peroxide in 3T3-L1 and C3H10T1/2 cells was inhibited by 50 μM febuxostat (Figure 4F,G). Given that ROS are produced in cells in response to exogenous stimuli, these results suggest that oxidative stress plays an important role in adipocyte differentiation in a manner that is inhibited by febuxostat.

### 3.4. Nrf2 Plays an Important Role in Adipocyte Differentiation

We further investigated the mechanisms underlying febuxostat-induced suppression of adipocyte differentiation. As ROS plays an important role in adipocyte differentiation, we focused on Nrf2, a transcription factor for various cytoprotective genes against oxidative stress. The Nrf2 gene was silenced with shRNA in 3T3-L1 and C3H10T1/2 cells, and the shRNA with the best silencing efficiency was selected (Appendix A). When well-silenced cells were cultured in adipogenic media, adipocyte differentiation with the appearance of Oil Red O-positive lipid droplets was markedly enhanced (Figure 5A,B). Conversely, the addition of the Nrf2 activator omaveloxolone inhibited the adipocytic differentiation of 3T3-L1 and C3H10T1/2 cells (Figure 5C,D). These results suggest that Nrf2 is a negative regulator of adipocyte differentiation under the present experimental conditions.

### 3.5. Febuxostat Promptly Induces Nuclear Translocation of Nrf2

Our results suggest that the inhibition of XO by febuxostat suppresses adipocyte differentiation by inhibiting XO-mediated ROS production. However, the relationship between febuxostat and Nrf2 activation has not been studied. Therefore, we examined the nuclear localization of Nrf2 in 3T3-L1 and C3H10T1/2 cells after treatment with febuxostat. The addition of omaveloxolone immediately increased the nuclear localization of Nrf2 within 1 h in 3T3-L1 and C3H10T1/2 cells (Figure 6A,B). Notably, the addition of febuxostat similarly promoted the nuclear localization of Nrf2 within 1 h, followed by the upregulation of Nrf2 target antioxidant genes, including *Ho-1* and *Nqo1*, in 3T3-L1 and C3H10T1/2 cells (Figure 6C,D), suggesting that febuxostat has potential as an Nrf2 activator. Intriguingly, Nrf2 activation with nuclear translocation was not induced by allopurinol (Figure 6E), another XO inhibitor. Therefore, febuxostat-induced Nrf2 activation appears to be independent of the XOR pathway.

### 3.6. Febuxostat Facilitates the Degradation of Keap1

Keap1 contains several highly reactive cysteine residues, and classic Nrf2 activators electrophilically modify the cysteine residues and thereby induce conformational changes in Keap1 to release Nrf2, leading to the nuclear localization of Nrf2. Such electrophilic modifications have also been demonstrated to accelerate the degradation of Keap1 by autophagy [31,32,33]. To investigate the mechanism underlying the induction of nuclear translocation of Nrf2 by febuxostat, we first examined the role of omaveloxolone in Keap1 degradation. Keap1 protein levels decreased over time upon inhibition of protein synthesis by cycloheximide, an inhibitor of translation, in 3T3-L1 and C3H10T1/2 cells (Figure 7A). The addition of omaveloxolone resulted in faster degradation of the Keap1 protein. Interestingly, the addition of febuxostat accelerated the degradation of the Keap1 protein (Figure 7B). Furthermore, the induction of Nrf2 nuclear translocalization, and thereby its target gene expression, and the facilitation of Keap1 degradation were similarly observed in MC3T3-E1 pre-osteoblastic cells (Figure 8), which further confirms the potential of febuxostat as an NRF2 activator. These results suggested that febuxostat can inactivate Keap1 to release and stabilize Nrf2 through conformational changes to facilitate its degradation.

## 4. Discussion

The present study demonstrated that febuxostat suppresses the accumulation of lipid droplets in adipocytes in vitro and alleviates the increase in adipose tissue mass in obese mouse models with a high-fat diet or ovariectomy. Regulation of adipocyte differentiation by targeting XO is expected to inhibit obesity due to menopause or overeating. XOR acts as a novel regulator of adipogenesis [30]. XOR is upregulated downstream of the transcription factor C/EBPβ expressed at a very early stage of adipocytic differentiation, which induces the expression of the transcription factors C/EBPα and PPARγ for further adipocytic maturation, and *Xor* gene silencing inhibits the expression of C/EBPα and PPARγ to abolish adipogenesis [30]. Consistent with the observation that XOR is upregulated downstream of C/EBPβ to induce the expression of C/EBPα and PPARγ, febuxostat was able to suppress the expression of C/EBPα and PPARγ, and thereby adipogenesis.

XOR generates xanthine dehydrogenase, which is converted into XO and exerts multiple biological activities, including dehydrogenase and oxidase activities [34]. In addition to purine catabolism, XO catalyzes the monovalent and divalent electron transfer to O_2_, which generates superoxide ions (O_2_^•−^) and hydrogen peroxide, respectively, thus producing ROS. Therefore, XOR can physiologically and pathologically regulate relevant signaling in part through ROS generation by XO [17]. NAC, an ROS scavenger, suppressed adipocyte differentiation in adipogenic media with IBMX, which is widely used to stimulate in vitro adipogenesis. IBMX is a nonspecific inhibitor of phosphodiesterase, which enhances intracellular cAMP levels [21,22]. Conversely, the addition of hydrogen peroxide further promoted adipocyte differentiation upon stimulation with an adipogenic medium. These results suggest a critical role for ROS in adipocyte differentiation and alleviation of adipocyte differentiation by suppressing XO-mediated ROS production using febuxostat.

Nuclear translocation of Nrf2 occurs parallel to excess ROS production [26,27]; however, febuxostat rapidly induced the nuclear translocation of Nrf2 in adipocytic and osteoblastic cells, similar to the Nrf2 activator omaveloxolone (Figure 6 and Figure 8). Intriguingly, Nrf2 nuclear translocalization was not induced by allopurinol (Figure 6E), another XO inhibitor, suggesting that the induction of Nrf2 nuclear translocation by febuxostat was independent of XO inhibition. Omaveloxolone is a synthetic oleanane triterpenoid that binds to Keap1 to release and stabilize Nrf2 [35,36]. Similar to omaveloxolone, febuxostat promptly induced Nrf2 nuclear translocation and facilitated the degradation of Keap1 (Figure 6 and Figure 7), suggesting its role as an Nrf2 activator. Cysteine residues are present in the thiol-rich KEAP1 protein via oxidation or alkylation [37]. Many cysteine residues of KEAP1 are modified to release Nrf2 by electrophilic molecules, such as omaveloxolone, implying an electrophilic interaction of febuxostat with Keap1. However, the precise mechanisms underlying Nrf2 activation by febuxostat remain unclear.

In concert with the suppression of XOR-mediated adipogenic signaling and XO-mediated ROS production, febuxostat is suggested to be able to directly induce the nuclear translocation of Nrf2 and upregulate a variety of Nrf2-target antioxidant genes to further contain ROS during adipocyte differentiation. The present study also suggests that treatment with febuxostat can ameliorate alimentary or postmenopausal obesity. We previously reported that febuxostat effectively suppresses RANKL-induced ROS production and osteoclastogenesis and that oral febuxostat administration alleviates bone loss in ovariectomized mice as an osteoporosis model. Therefore, febuxostat may be a therapeutic option for postmenopausal obesity with osteoporosis. Given the wide range of regulatory roles of Nrf2, the therapeutic impact of febuxostat can be envisioned in a broad spectrum of physiological and pathological conditions, including skewed redox metabolism, proteostasis, inflammation, and cancers, which warrant further study. To our best knowledge, there has been no clinical study with febuxostat aiming to prevent obesity. However, febuxostat has been reported in clinical studies to have a direct ameliorating effect on oxidative stress in various pathological conditions, including hemodialysis patients with endothelial dysfunction [38] and patients with chronic heart failure and hyperuricemia [39].

There are several experimental limitations to the present study. We produced the data from young mouse models with a high-fat diet or ovariectomy but not from aged mice with low metabolic activity. Optimization of dosing schedules for febuxostat is needed. We mainly analyzed subcutaneous adipose tissues in vivo, but we should also study the fat accumulation in different organs, including the liver and bone marrow. As the next step in our study, we plan to study the effects of febuxostat on adiposity in the bone marrow and liver in these mouse models. Most importantly, the anti-obesity effects of febuxostat should be determined in human subjects. The present results may warrant clinical studies with febuxostat to prevent obesity due to menopause or overeating.

## 5. Conclusions

Febuxostat alleviates obesity in mouse models with a high-fat diet or ovariectomy. Oxidative stress stimulates in vitro adipogenesis, which is disrupted by febuxostat. Febuxostat not only suppresses xanthine oxidase-mediated ROS production but also rapidly induces nuclear translocation of Nrf2 with antioxidant upregulation to ameliorate ROS-induced adipogenesis.

## Figures and Tables

**Figure 1 antioxidants-12-00133-f001:**
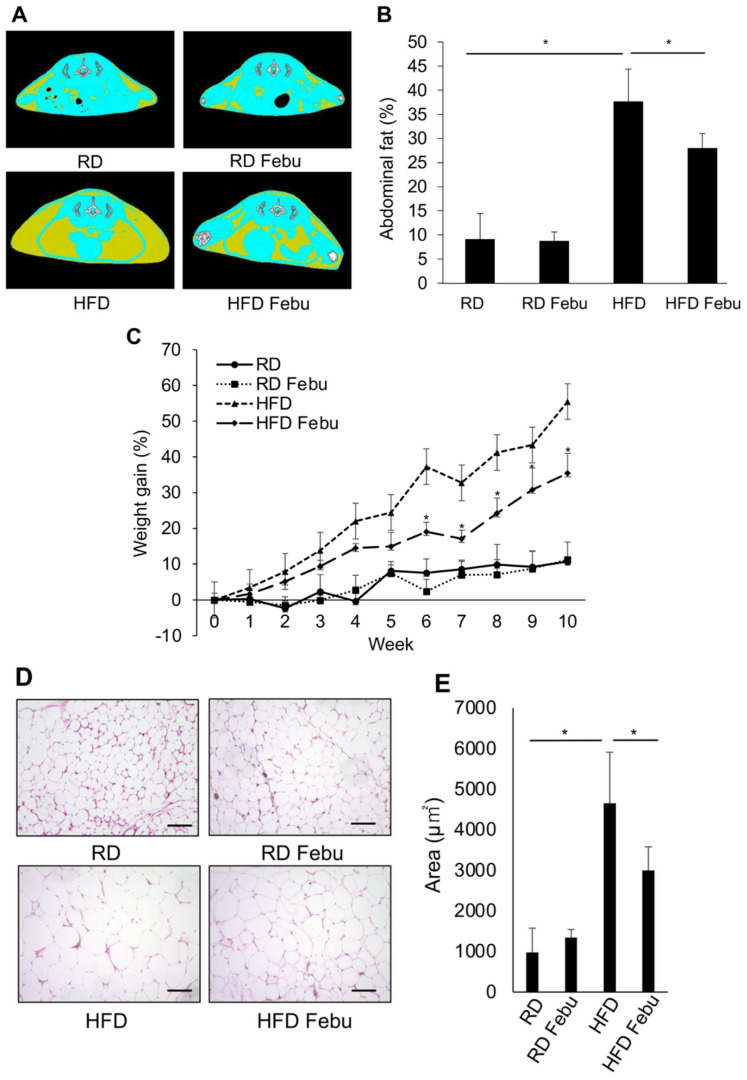
Febuxostat (Febu) ameliorates obesity in high-fat diet-fed (HFD) mice. Regular diet (RD) or high-fat diet (HFD) fed mice were treated for 10 weeks with or without oral administration of Febu. (**A**) Representative micro-CT images of RD and HFD mice with or without Febu treatment are shown. Yellow, adipose tissue; blue, lean mass; white, vertebral bone. (**B**) Quantitative analysis of abdominal fat volume from the level of 1st to 6th lumbar vertebrae using micro-CT images at 10 weeks. *n* = 5 per group. Results are expressed as the mean ± SD. * *p* < 0.05. (**C**) Percent increase in body weight from baseline in RD and HFD mice with or without Febu treatment is shown. *n* = 5 per group. Results are expressed as the mean ± SD. * *p* < 0.05 vs. HFD at the indicated time point. (**D**) Representative images of hematoxylin and eosin (H&E) staining of the visceral white adipose tissue. Original magnification, ×200. Bar, 100 μm. (**E**) The area of adipocytes was measured. *n* = 5 per group. Results are expressed as the mean ± SD. * *p* < 0.05.

**Figure 2 antioxidants-12-00133-f002:**
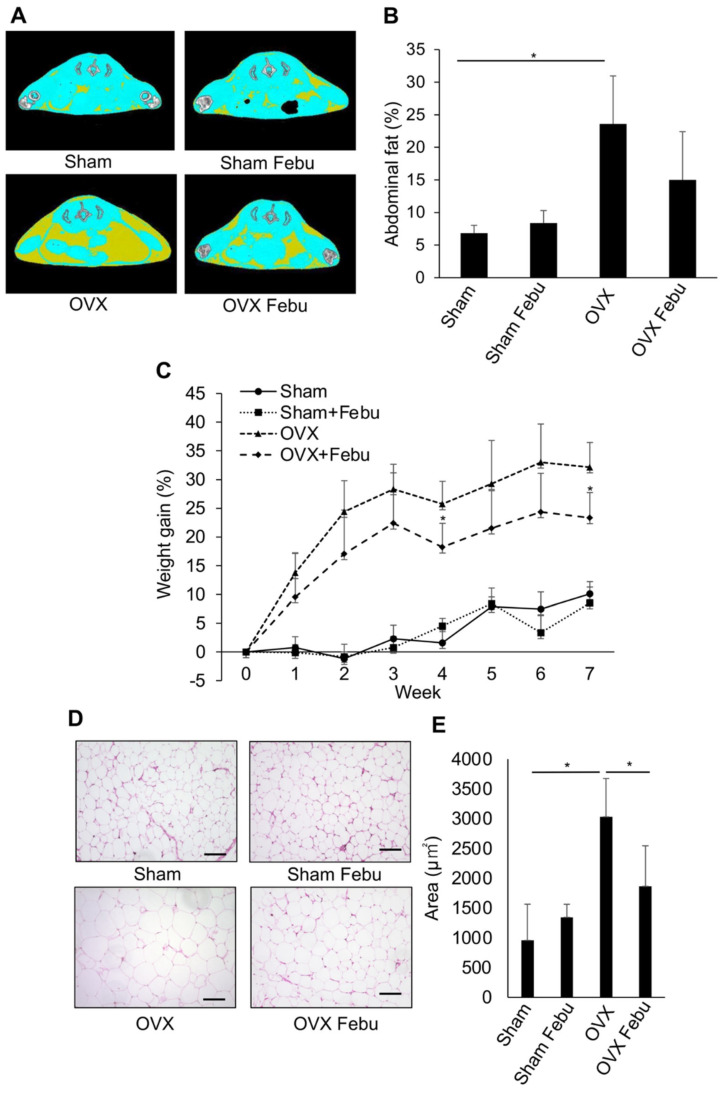
Febuxostat ameliorates obesity in ovariectomized (OVX) mice. Sham-operated or ovariectomized (OVX) mice were treated for seven weeks with or without oral administration of Febu. (**A**) Representative micro-CT images of sham-operated and OVX mice with or without Febu treatment are shown. (**B**) Quantitative analysis of the abdominal fat from 1st to 6th lumbar vertebrae. *n* = 5 per group. Results are expressed as the mean ± SD. * *p* < 0.05. (**C**) Percent increase in body weight from the baseline in sham-operated and OVX mice with or without Febu treatment is shown. *n* = 5 per group. Results are expressed as the mean ± SD. * *p* < 0.05. vs. OVX at the indicated time point. (**D**) Representative images of H&E staining of the visceral white adipose tissue. Original magnification, ×200. Bar, 100 μm. (**E**) The area of adipocytes was measured. *n* = 5 per group. Results are expressed as the mean ± SD. * *p* < 0.05.

**Figure 3 antioxidants-12-00133-f003:**
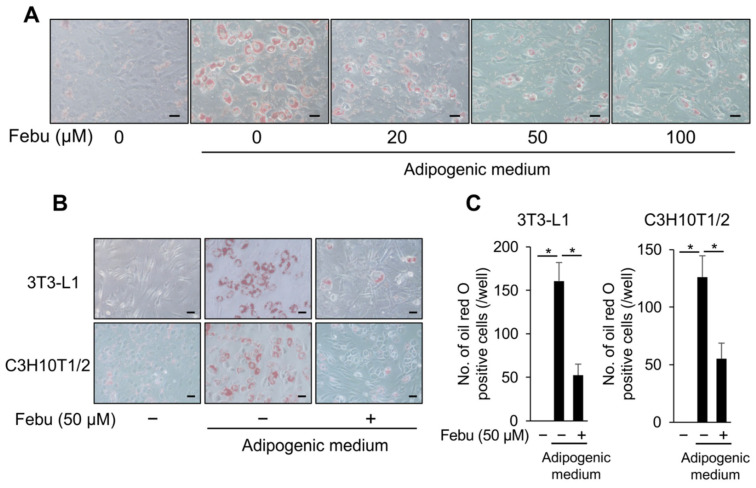
Febuxostat inhibits adipocyte differentiation. (**A**) C3H10T1/2 cells were cultured in adipocyte differentiation medium for seven days. Febu was added at the indicated concentrations. Representative image of Oil Red O staining. Original magnification, ×200. Bar, 100 μm. (**B**) 3T3-L1 and C3H10T1/2 cells were cultured in adipocyte differentiation medium with or without Febu for seven days. Representative image of Oil Red O staining. Original magnification, ×200. Bar, 100 μm. (**C**) Oil Red O-positive cells were counted. Data are expressed as the mean ± SD (*n* = 4). * *p* < 0.05. (**D**) 3T3-L1 and C3H10T1/2 cells were seeded in culture dishes, and Febu was added at the indicated concentrations. Cell viability was measured using the 2-(2-methoxy-4-nitrophenyl)-3-(4-nitrophenyl)-5-(2,4-disulphophenyl)-2*H*-tetrazolium monosodium salt (WST-8) assay. Data are expressed as the mean ± SD. (**E**) Expression of adipogenic differentiation markers in 3T3-L1 cells in the presence or absence of Febu analyzed by western blotting. β-actin served as the loading control. Relative changes in the band intensities normalized by respective loading controls are indicated. (**F**) mRNA expression of adipogenic differentiation markers in 3T3-L1 and C3H10T1/2 cells in the presence or absence of Febu using real-time PCR. *Gapdh* served as an endogenous control to normalize each sample. Data are expressed as the mean ± SD (*n* = 3). * *p* < 0.05.

**Figure 4 antioxidants-12-00133-f004:**
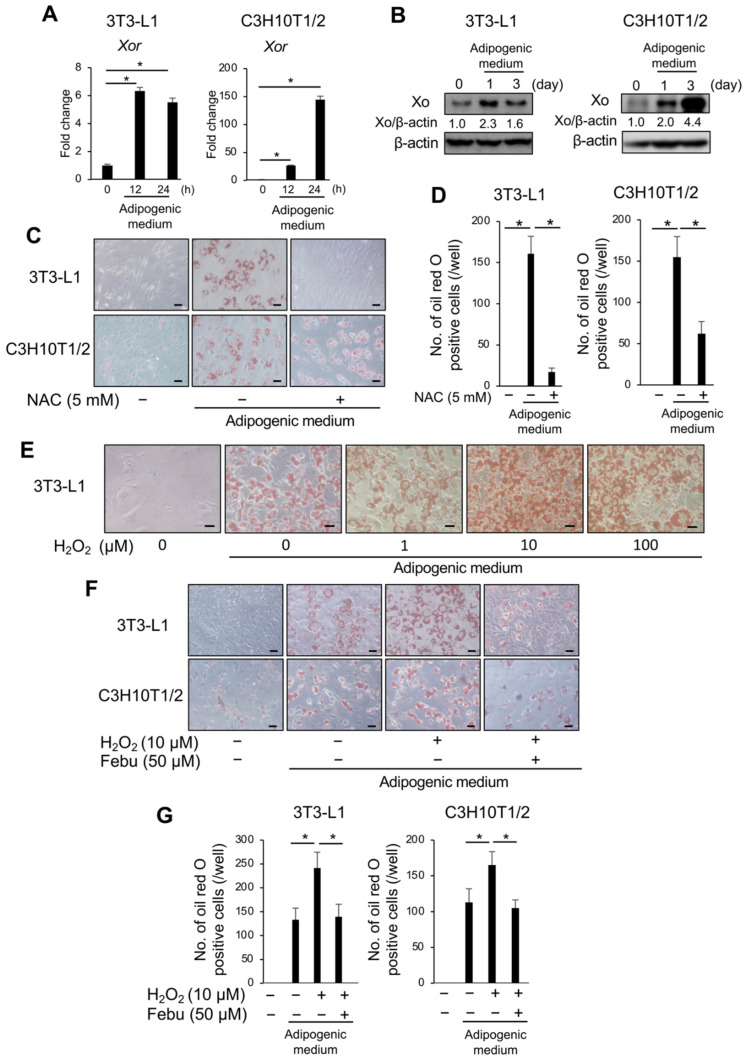
Xanthine oxidase (XO) is induced during adipogenesis, and Febuxostat inhibits ROS-induced excessive adipocyte differentiation. (**A**) 3T3-L1 and C3H10T1/2 cells were cultured in adipocyte differentiation medium. *Xor* mRNA expression was determined using real-time PCR. *Gapdh* served as an endogenous control to normalize each sample. Data are expressed as the mean ± SD (*n* = 3). * *p* < 0.05. (**B**) Xor protein level was analyzed by western blotting. β-actin served as a loading control. Relative changes in the band intensities normalized by respective loading controls are indicated. (**C**) 3T3-L1 and C3H10T1/2 cells were cultured in an adipocyte differentiation medium with or without *N*-Acetyl-l-cysteine (NAC) for seven days. Representative image of Oil Red O staining. Original magnification, ×200. Bar, 100 μm. (**D**) Oil Red O-positive cells were counted. Data are expressed as the mean ± SD (*n* = 4). * *p* < 0.05. (**E**) 3T3-L1 cells were cultured in an adipocyte differentiation medium with the indicated concentration of hydrogen peroxide (H_2_O_2_) for 48 h. The cells were then washed and further cultured for seven days in the differentiation medium without hydrogen peroxide. Representative image of Oil Red O staining. Original magnification, ×200. Bar, 100 μm. (**F**) 3T3-L1 and C3H10T1/2 cells were cultured in an adipocyte differentiation medium in the presence or absence of hydrogen peroxide, with or without Febu for 48 h. The cells were then washed and further cultured for seven days in the differentiation medium without hydrogen peroxide with or without Feb. Representative image of Oil Red O staining. Original magnification, ×200. Bar, 100 μm. (**G**) Oil Red O-positive cells were counted. Data are expressed as the mean ± SD (*n* = 4). * *p* < 0.05.

**Figure 5 antioxidants-12-00133-f005:**
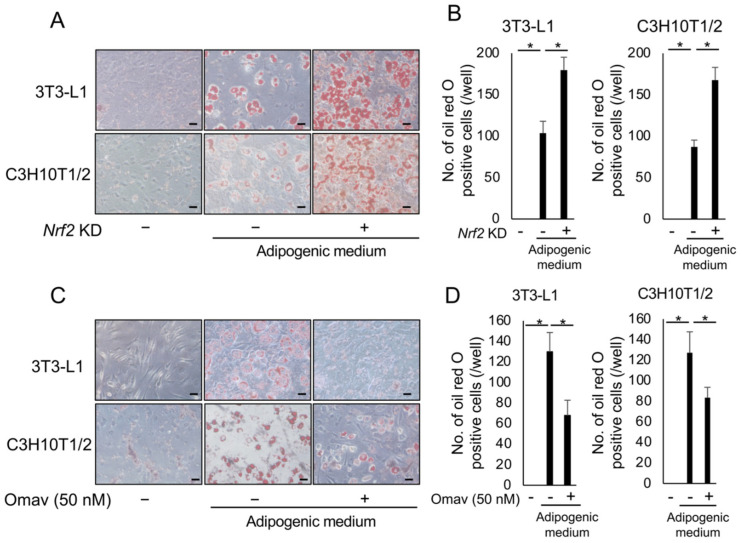
Nuclear factor erythroid 2-related factor 2 (Nrf2) plays an important role in adipocyte differentiation. (**A**) Nrf2 knockdown 3T3-L1 and C3H10T1/2 cells were cultured in an adipocyte differentiation medium for seven days. Representative image of Oil Red O staining. Original magnification, ×200. Bar, 100 μm. (**B**) Oil Red O-positive cells were counted. Data are expressed as the mean ± SD (*n* = 4). * *p* < 0.05. (**C**) 3T3-L1 and C3H10T1/2 cells were cultured in an adipocyte differentiation medium with or without omaveloxolone (Omav, a Nrf2 activator) (50 nM) for seven days. Representative image of Oil Red O staining. Original magnification, ×200. Bar, 100 μm. (**D**) Oil Red O-positive cells were counted. Data are expressed as the mean ± SD (*n* = 4). * *p* < 0.05.

**Figure 6 antioxidants-12-00133-f006:**
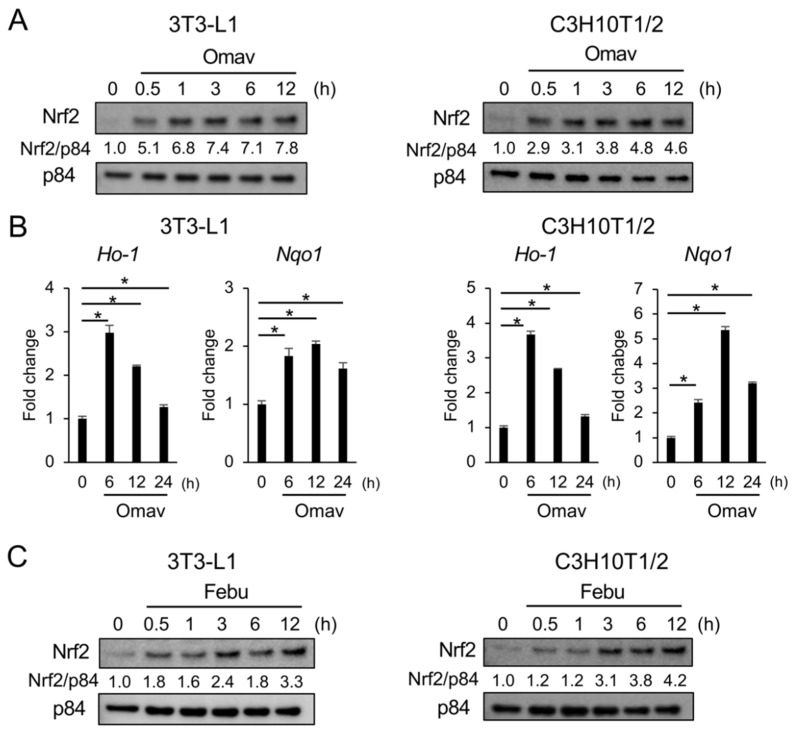
Febuxostat facilitates nuclear translocation of Nrf2. (**A**,**C**) 3T3-L1 and C3H10T1/2 cells were cultured with Omav or Febu for the indicated time period. Nrf2 levels in the nucleus were analyzed by western blotting. p84 served as a loading control. Relative changes in the band intensities normalized by respective loading controls are indicated. (**B**,**D**) mRNA expression of *Ho-1* and *Nqo1*, the target genes of Nrf2, was determined using real-time PCR. *Gapdh* served as an endogenous control to normalize each sample. Data are expressed as the mean ± SD (*n* = 3). * *p* < 0.05. (**E**) 3T3-L1 and C3H10T1/2 cells were cultured with allopurinol (Allo) for the indicated time period. Nrf2 levels in the nucleus were analyzed by western blotting, and p84 served as a loading control. Relative changes in the band intensities normalized by respective loading controls are indicated.

**Figure 7 antioxidants-12-00133-f007:**
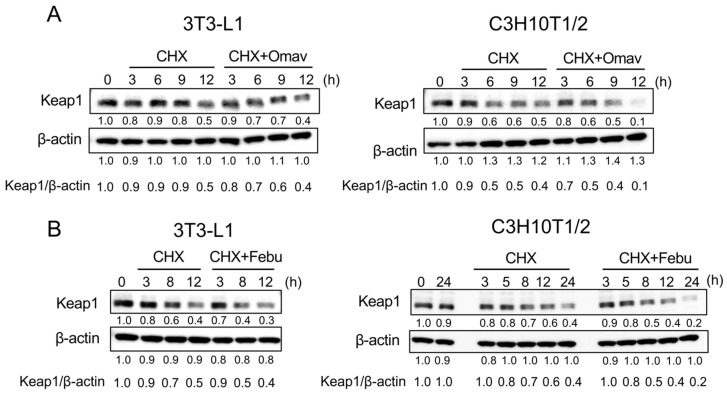
Febuxostat facilitates the degradation of kelch-like ECH-associated protein 1 (Keap1). (**A**,**B**) 3T3-L1 and C3H10T1/2 cells were cultured in the presence of cycloheximide with or without Omav or Febu for the indicated time period. Keap1 levels were analyzed by western blotting. β-actin served as a loading control. Relative changes in the band intensities normalized by respective loading controls are indicated.

**Figure 8 antioxidants-12-00133-f008:**
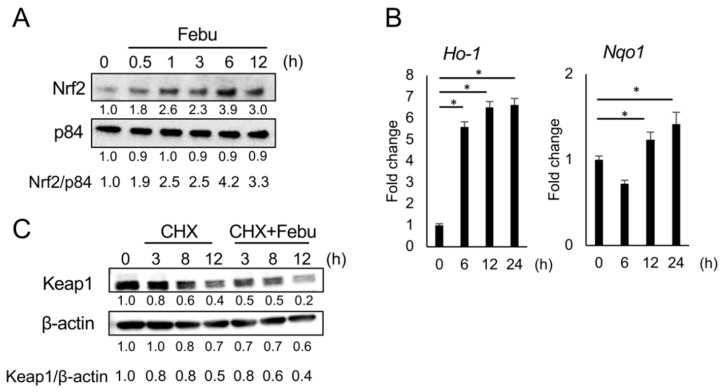
Febuxostat facilitates nuclear translocation of Nrf2 and degradation of Keap1 in MC3T3-E1 pre-osteoblastic cells. (**A**) MC3T3-E1 cells were cultured with Febu for the indicated time period. Nrf2 levels in the nucleus were analyzed by western blotting. p84 served as a loading control. Relative changes in the band intensities normalized by respective loading controls are indicated. (**B**) mRNA expression of *Ho-1* and *Nqo1*, the target genes of Nrf2, was determined using real-time PCR. *Gapdh* served as an endogenous control to normalize each sample. Data are expressed as the mean ± SD (*n* = 3). * *p* < 0.05. (**C**) MC3T3-E1 cells were cultured in the presence of cycloheximide with or without Febu for the indicated time period. Keap1 levels were analyzed by western blotting. β-actin served as a loading control. Relative changes in the band intensities normalized by respective loading controls are indicated.

## Data Availability

The data presented in this study are available on request to the corresponding authors.

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
