# Peer review of "The Xanthine Oxidase Inhibitor Febuxostat Suppresses Adipogenesis and Activates Nrf2"

_antioxidants, 2023, doi:10.3390/antiox12010133_

Round 1
Reviewer 1 Report
The manuscript presents interesting data demonstrating that febuxostat, an inhibitor of xanthine oxidase, alleviates the increase in adipose tissue mass in obese mouse induced by a high-fat diet or ovariectomy and disrupts in vitro adipocytic differentiation in adipogenic media, at least partly at least in part by suppressing ROS production and Nrf2 activation.
Results of the study point to a perspective of use of febuxostat for preventing of obesity induced by menopause or overeating.
Remarks
Does the anti-obesity effect due to inhibition of xanthine oxidase or rather to Nrf2 activation? The second possibility would suggest modification of the last sentence of the abstract.
Was the level of ROS or other indices of oxidative stress measured in the study?
Fig. 4. “3T3-L1 cells were cultured in an adipocyte differentiation medium with the indicated concentration of hydrogen peroxide (H2O2) for seven days”. How was it done? Was hydrogen peroxide given in a single bolus or was its concentration somehow maintained during incubation? Hydrogen peroxide is decomposed by cells in culture.
Fig. 7 and 8. Results of densitometric evaluation of the blots should be provided.
RANKL, please explain the acronym on the first use.
Reviewer 2 Report
I’ve read with attention the paper of Higa et al. that is potentially of interest. The background and aim of the study have been clearly defined. The methodology applied is overall correct, the results are reliable and adequately discussed. I’ve only some minor comments:
- Tha abstract should contain the main 2-3 quantitative data
- The authors report to have used t-test and ANOVA, but without having verified if the data distribution was normal. If it is, it should be corrected
- If the hypothesis of the authors is true, we should have observed some effect of febuxostat on anthropometric parameters in humans as well, especially considering the large number of patients treated with this drug and of patients involved in RCTs. Is there any clinical evidence of it? If yes, it shoudl be cited, if not, it should be discussed
- The authors should shortly discuss the potential experimental limitation of their approach and suggest the next step of their research
